# Non-Pessimistic Predictions of the Distributions and Suitability of *Metasequoia glyptostroboides* under Climate Change Using a Random Forest Model

**Xiaoyan Zhang** [1,2], **Haiyan Wei** [2,*], **Xuhui Zhang** [1,2], **Jing Liu** [1,2], **Quanzhong Zhang** [1,2] **and Wei Gu** [1,3,*]

[1] National Engineering Laboratory for Resource Development of Endangered Crude Drugs in Northwest China, Shaanxi Normal University, Xi'an 710119, China; zhangxiaoyan@snnu.edu.cn (X.Z.); 18592030436@snnu.edu.cn (X.Z.); lhlj@snnu.edu.cn (J.L.); zhangqz@snnu.edu.cn (Q.Z.)
[2] School of Geography and Tourism, Shaanxi Normal University, Xi'an 710119, China
[3] College of Life Sciences, Shaanxi Normal University, Xi'an 710119, China
[*] Correspondence: weihy@snnu.edu.cn (H.W.); weigu@snnu.edu.cn (W.G.); Tel.: +86-29-8531-0525 (H.W.); +86-29-8531-0266 (W.G.)

**Abstract:** *Metasequoia glyptostroboides* Hu & W. C. Cheng, which is a remarkable rare relict plant, has gradually been reduced to its current narrow range due to climate change. Understanding the comprehensive distribution of *M. glyptostroboides* under climate change on a large spatio-temporal scale is of great significance for determining its forest adaptation. In this study, based on 394 occurrence data and 10 bioclimatic variables, the global potential distribution of *M. glyptostroboides* under eight different climate scenarios (i.e., the past three, the current one, and the next four) from the Quaternary glacial to the future was simulated by a random forest model built with the biomod2 package. The key bioclimatic variables affecting the distribution of *M. glyptostroboides* are BIO2 (mean diurnal range), BIO1 (annual mean temperature), BIO9 (mean temperature of driest quarter), BIO6 (min temperature of coldest month), and BIO18 (precipitation of warmest quarter). The result indicates that the temperature affects the potential distribution of *M. glyptostroboides* more than the precipitation. A visualization of the results revealed that the current relatively suitable habitats of *M. glyptostroboides* are mainly distributed in East Asia and Western Europe, with a total area of approximately $6.857 \times 10^6$ km$^2$. With the intensification of global warming in the future, the potential distribution and the suitability of *M. glyptostroboides* have a relatively non-pessimistic trend. Whether under the mild (RCP4.5) and higher (RCP8.5) emission scenarios, the total area of suitable habitats will be wider than it is now by the 2070s, and the habitat suitability will increase to varying degrees within a wide spatial range. After speculating on the potential distribution of *M. glyptostroboides* in the past, the glacial refugia of *M. glyptostroboides* were inferred, and projections regarding the future conditions of these places are expected to be optimistic. In order to better protect the species, the locations of its priority protected areas and key protected areas, mainly in Western Europe and East Asia, were further identified. Our results will provide theoretical reference for the long-term management of *M. glyptostroboides*, and can be used as background information for the restoration of other endangered species in the future.

**Keywords:** climate change; *Metasequoia glyptostroboides* Hu & W. C. Cheng; random forest; potential distribution; glacial refugia; protected areas

## 1. Introduction

Climate is a crucial driver of physiological processes related to the species survival [1]. Changes in spatio-temporal climate patterns significantly influence both temperature and precipitation, which in turn affect the growth conditions and geographical distribution of species [2]. Many studies have confirmed that when hydrothermal conditions exceed the metabolic range of species, these species are either at the risk of extinction or migrate to the poles or upwards to adapt to the change in climate [3]. The Intergovernmental Panel on Climate Change (IPCC) estimates that temperatures could increase by 1.5 °C or more by 2030–2052 if global warming continues to increase at the current rate, which will have an impact on biodiversity and ecosystems [4]. If the dynamic nonlinear response between climate change and species distribution can be visualized, it will help mitigate any potential threat that climate change may bring to the species habitats [5].

The first species distribution models (SDMs) package called BIOCLIM was developed in the mid-1980s to study the effects of climate on species distributions [6]. SDMs, also known as ecological niche modeling (ENM), aim to predict species geographic distribution in projected range and threat levels through a set of statistical methods based on limited species records and corresponding environmental variables [7]. In recent years, with the development of computer, geo-information system (GIS), and remote sensing (RS) technology, SDMs have been developed rapidly. The existing SDMs can be divided into regression models, niche models, and machine learning models [8]. In past decades, SDMs have been of interest due to their widespread application in the exploration of biodiversity [9], the assessment of biological invasions [10], the research on biological productivity [11], and especially the prediction of species potential distributions [12]. Following the recent recommendations for pseudo-absent optimal models [13], many scholars believe that the predictive ability and overall performance of random forest (RF) models are optimal [14–16]. The RF model is an ensemble machine learning approach [17], which can build a large number of regression trees for classification and regression by selecting multiple sub-samples from the total data. This algorithm avoids the shortcomings of previous machine learning models that are prone to overfitting and has received increasing attention for the prediction of species potential distribution in recent years [18,19].

*Metasequoia glyptostroboides* Hu & W. C. Cheng, the dawn redwood, is a remarkable, rare relict plant heralded as a 'living fossil' [20], and was listed on the International Union for Conservation of Nature (IUCN) Red List of Threatened Species in 2013 [21]. It is an endangered deciduous conifer and the sole living species of the genus *Metasequoia*, which was once thought to have become extinct in the Miocene epoch [22]. The rediscovery of a live *M. glyptostroboides* specimen in Lichuan, Hubei province, China, in 1941, was one of the greatest scientific contributions to botany in nearly a century [23]. The natural occurrence of *M. glyptostroboides* was extremely limited, and almost all of the very extensive occurrences that are now outside China were introduced and propagated from the seed stock of China's Sichuan-Hubei border. As an important relict plant, *M. glyptostroboides* has considerable ornamental, medicinal, and ecological value. Because of its beautiful shape and soft material, the dawn redwood can be used in horticulture, construction, papermaking, and other industries. Moreover, its leaves and fruits have antipyretic and detoxifying, anti-inflammatory, and analgesic effects [24], and the volatile oil in its seeds contains various active ingredients with antibacterial effects [25]. Hence, the considerable medicinal value of *M. glyptostroboides* is also widely used in medicine, chemistry, pharmacology, and other fields. In the last few decades, the comprehensive distribution influenced by climate change of *M. glyptostroboides* has not been investigated, although the main research on it has involved the field of genetic diversity [26], physiological characteristics [27], and cultivation and management [28]. Therefore, it is important to quantify the impact of irreversible climate change on potential distribution and habitat suitability of *M. glyptostroboides* on a global scale.

Objectively, although *M. glyptostroboides* luckily survived glacial movement, the once prosperous *M. glyptostroboides* gradually degenerated into today's narrow distribution, and would face more complex ecological threats under future climate change [29]. Approximately 2 million years ago, the arrival of glaciation of the Quaternary led to transient and abrupt climate change, which considerably

forced the growth, evolution, and distribution of biology changes. Fortunately, many of the relict plants like *M. glyptostroboides* are thought to have withstood environmental upheavals and survived in the refugia, where the different microclimates can provide suitable conditions for species to survive in both warm and cold periods [30]. After the rediscovery of *M. glyptostroboides*, this kind of relict plant quickly attracted unprecedented attention and protection for half a century [31]. However, even if individual trees are protected, many habitats of *M. glyptostroboides* are not, which makes natural regeneration difficult [29]. For several years, inferring the glacial refugia of relict plants by combining paleobotany, palynology, systematic geography, and molecular biology has become a research hotspot [32–35]. Many scholars believe that refugia theory has special value in explaining both the resurrection of relict plants and the persistence of biodiversity [36–38]. Therefore, the ecological comprehensive value of *M. glyptostroboides* can be focused on by determining its historically geographical scope and its refugia. In the context of climate change, whether the suitability of the habitat (especially refugia) of *M. glyptostroboides* will be affected by future global warming is also one of our concerns. Thus, it is necessary and meaningful to determine the potential geographic distribution of this species and predict how climate change will affect its geographic scope.

Based on species records and high-resolution bioclimatic variables of different climate conditions, we used the RF model to evaluate the global potential distribution and habitat suitability of *M. glyptostroboides*. The objectives of this study are (1) to estimate the potential distribution of *M. glyptostroboides* under current, past, and future climatic scenarios; (2) to infer the location and speculate the impact of future climate change on glacial refugia for *M. glyptostroboides* during the Quaternary glaciar; and (3) to forecast the change of habitat suitability and identify protected areas of *M. glyptostroboides* for further research. The results will provide theoretical reference and reasonable suggestion for the protection, management and cultivation of *M. glyptostroboides* in the future, so as to explore enlightenment from nature.

## 2. Materials and Methods

### 2.1. Species Records

A database of occurrences was constructed using four main sources: the Global Biodiversity Information Facility (GBIF, http://www.gbif.org/), the Atlas of Living Australia (ALA, https://www.ala.org.au/), the Chinese Virtual Herbarium (CVH, http://www.cvh.org.cn/), and the published literature. In total, 853 records with longitude and latitude information were obtained by filtering the data before 1950. Finally, 394 sampling points (Figure 1) were selected for this study after removing the over-fit records and the records with unclear spatial information. We converted species records into a 0.5-degree spatial resolution based on the WGS-84 spatial coordinate system to reduce the error caused by different data sources.

According to the visualization, the global distributions of *M. glyptostroboides* are mainly distributed in the mid-latitudes of the Northern Hemisphere. The occurrences used for analysis included major centers in Europe (156 records) and Asia (169 records), and outliers in North America, South Africa, and Australia (a total of 69 records). Inspections of these sites indicate that there are few records from the botanical gardens.

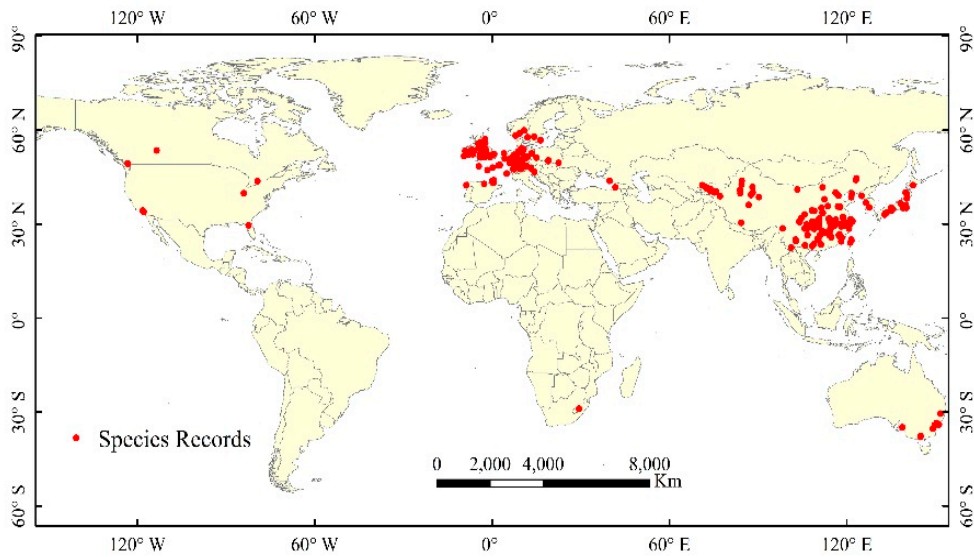

**Figure 1.** The global sampling points of *M. glyptostroboides*.

## 2.2. Bioclimatic Variables

We collected 19 bioclimatic variables with a high spatial resolution of 1 km$^2$ (Table 1) to represent annual (e.g., BIO1 and BIO12) and seasonal (e.g., BIO4 and BIO15) trends as well as extreme or limiting environmental factors (e.g., BIO5 and BIO6, and BIO16 and BIO17, respectively) in this study. The data layers, which are available from the WorldClim database (WorldClim Version 1.4, http://www.worldclim.org/), were interpolated from the average monthly climate data from weather stations. Eliminating the autocorrelation among predictors helps to avoid the prediction error caused by multicollinearity of bioclimatic factors. Therefore, Pearson correlation coefficients (*r*) of 19 bioclimatic variables were calculated, and only one out of each strongly correlated factor group (*r* > 0.8) was retained. After screening, 10 bioclimate variables were selected to participate in the modeling (Table 1).

**Table 1.** Overview of bioclimatic variables used in this study.

| Abbreviation | Parameter (Unit) | Selected |
|---|---|---|
| BIO1 | Annual Mean Temperature (°C) | √ |
| BIO2 | Mean Diurnal Range [Mean of monthly (max temp–min temp)] (°C) | √ |
| BIO3 | Isothermality (BIO2/BIO7) (*100) (%) | |
| BIO4 | Temperature Seasonality (standard deviation *100) (°C) | |
| BIO5 | Max Temperature of Warmest Month (°C) | |
| BIO6 | Min Temperature of Coldest Month (°C) | √ |
| BIO7 | Temperature Annual Range (BIO5–BIO6) (°C) | √ |
| BIO8 | Mean Temperature of Wettest Quarter (°C) | |
| BIO9 | Mean Temperature of Driest Quarter (°C) | √ |
| BIO10 | Mean Temperature of Warmest Quarter (°C) | √ |
| BIO11 | Mean Temperature of Coldest Quarter (°C) | |
| BIO12 | Annual Precipitation (mm) | √ |
| BIO13 | Precipitation of Wettest Month (mm) | √ |
| BIO14 | Precipitation of Driest Month (mm) | |
| BIO15 | Precipitation Seasonality (Coefficient of Variation: mean/SD*100) (%) | |
| BIO16 | Precipitation of Wettest Quarter (mm) | √ |
| BIO17 | Precipitation of Driest Quarter (mm) | |
| BIO18 | Precipitation of Warmest Quarter (mm) | √ |
| BIO19 | Precipitation of Coldest Quarter (mm) | |

Bioclimate data from different scenarios (current one, past three, and future four) were applied to predict the potential distribution of *M. glyptostroboides* on a large space-time scale. Current climate

scenario, as a baseline climate scenario, was generated by the Kriging interpolation of observed data, representative of 1960–1990. Future and past climate scenarios were usually generated by Global Climate Models (GCMs, also known as General Circulation Models), which have been downscaled and calibrated (bias corrected) by WorldClim. The paleoclimate data, whose original data were made available by Coupled Model Intercomparison Project (CMIP5), of which the Last Interglacial (LIG, about 130–110 ka BP, which was similar to the current climate), the Last Glacial Maximum (LGM, about 22 ka BP lower than the current temperature), and the Mid-Holocene (MH, about 6 ka BP higher than the current temperature) were selected. The future bioclimate data come from the climate projections from the Community Climate System Model 4 (CCSM4, a most recent GCM climate projection). Scientific assumptions about future climatic scenarios were produced by the Fifth Assessment Report of the IPCC for the new generation of emission scenarios, known as 'representative concentration pathways' (RCPs). The RCPs were identified based on the possible trajectories of greenhouse gas emissions from all sources relative to the pre-industrial period. The description of four kinds of RCPs (RCP2.6, RCP4.5, RCP6.0, and RCP8.5) can be found at http://www.pik-potsdam.de/--mmalte/rcps/. In this study, both mild (RCP4.5) and severe (RCP8.5) emission scenarios were selected for the 2050s (average across 2041–2060) and 2070s (average across 2061–2080), respectively. With this method, two greenhouse gas scenarios in two periods were combined into four combinations to represent future climate conditions.

## 2.3. RF Model Building and Validation

In this study, RF model provided by the biomod2 package developed in R environment (https://www.r-project.org/) was used for prediction. Based on 394 species records of *M. glyptostroboides* and 10 bioclimate variables, RF model was selected to predict the potential distribution maps of *M. glyptostroboides* in the eight climate scenarios. To improve the model capacity, the presence data were formatted using the 'random' algorithm to randomly generate 2000 pseudo-presence points for every model, and divided the species data into two groups (80% for the training set and 20% for the testing set). The models were cross-validated in four replicate runs of each model to minimize the errors and ensure we obtained more realistic predictions.

We used the true skill statistic (TSS), Cohen's KAPPA (KAPPA), and receiver operating characteristic (ROC) curves to evaluate each model, which are considered to be the most popular dimensionless indicators for verifying the accuracy of SDM [39]. ROC is defined as the test's true-positive rate (sensitivity) plotted against its false-positive rate (1-specificity), and is used to assess the model's ability by distinguishing between distributed and non-distributed clustering [40]. Kappa can be used to evaluate the consistency between the sample data and the simulated results [41]. TSS is a threshold-dependent evaluator (equal to sensitivity + specificity − 1), which not only retains the advantages of kappa, but also corrects the disadvantages of kappa's susceptibility to species distribution. The thresholds of the three evaluation indexes are from 0 to 1, and the higher value indicates the higher performance of the models. Usually, if ROC is greater than 0.9, TSS is greater than 0.85, or kappa is greater than 0.8, then the model process is considered to perform well [42].

## 2.4. Data Analyses of Key Bioclimatic Variables and Habitat Suitability

The weight of bioclimatic variables affecting the growth and the distribution of *M. glyptostroboides* can be obtained after modeling. Moreover, corresponding statistical analyses were carried out to identify the bioclimatic variables with a cumulative contribution rate exceeding 80% (in descending order of weight) in each model. We then selected the key bioclimatic variables that exist in all climate scenarios based on these bioclimatic variables.

The habitat suitability index (HSI) is widely used in species habitat evaluation and can be obtained from the model outputs [43–46]. For further analyses, HSI was classified into four levels of habitat suitability [47]: unsuitable (HSI < 0.3), marginally suitable (0.3 ≤ HSI < 0.5), moderately suitable (0.5 ≤ HSI < 0.7), and highly suitable (HSI ≥ 0.7).

### 2.5. Identification of Glacial Refugia and Protected Areas

In combination with previous studies on relict plants, glacial refugia can be inferred by superimposing the species potential distribution under both the paleoclimate and the current climate [48]. In this study, we hypothesized that the evolution of *M. glyptostroboides* is historic and continuous, and inferred glacial refugia by coupling the potential distribution of *M. glyptostroboides* in the LGM (before the Quaternary) and in the current. The area is defined as a 'refugia' if the area is suitable for *M. glyptostroboides* under both climate scenarios. On the other hand, habitat adaptation level trends in future climate scenarios were calculated and used to identify protected areas. We defined 'priority protected areas' as the areas where the suitability level of *M. glyptostroboides* is likely to decline in any future climate scenarios, and define 'key protected areas' as the areas where the suitability levels will increase in all future climate scenarios.

## 3. Results

### 3.1. Model Reliability and Key Bioclimatic Variables

The eight models in this study are of high reliability, with the ROC value of 0.984 ± 0.009 (Figure 2). Additionally, the TSS is 0.885 ± 0.033, and the KAPPA is 0.839 ± 0.054. Thus, the model results could be considered to be satisfactory.

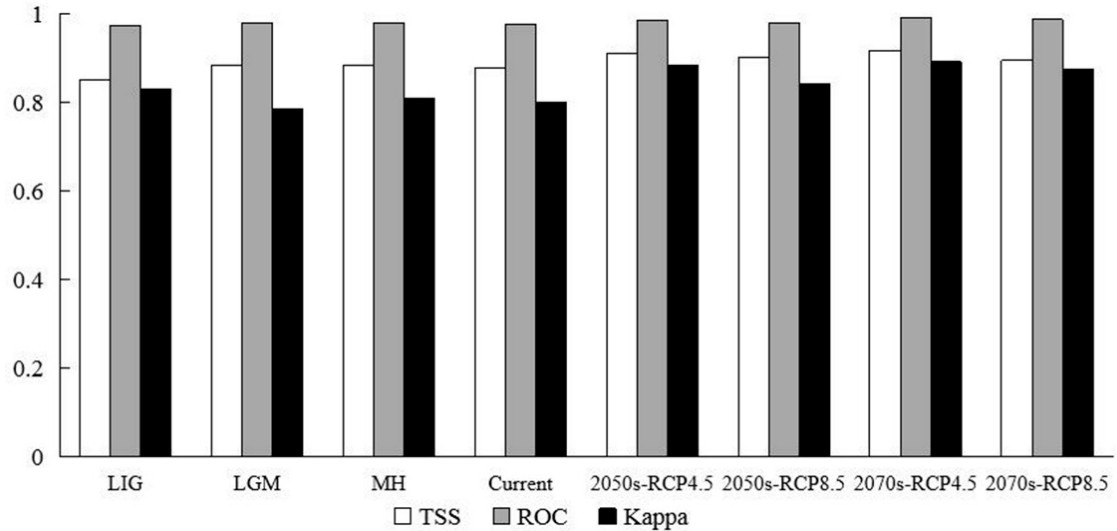

**Figure 2.** Evaluation indexes of our model under eight climate scenarios. LIG represents the Last Interglacial, LGM represents the Last Glacial Maximum, MH represents the Mid-Holocene, Current represents the current scenarios, 2050s-RCP4.5 represents the RCP4.5 in the 2050s, 2050s-RCP8.5 represents the RCP8.5 in the 2050s, 2070s-RCP4.5 represents the RCP 4.5 in the 2070s, and 2070s-RCP8.5 represents the RCP 8.5 in the 2070s, respectively.

Bioclimatic variables affecting the growth and the distribution of *M. glyptostroboides* can be obtained by the model. Five to seven bioclimatic variables accounting for the top 80% of the cumulative contribution rate were obtained in each model (Figure 3a). Based on further statistics, five of the bioclimatic variables were singled out as key bioclimatic variables (Figure 3a, colored portion), of which four are related to temperature, and one is related to precipitation. They are BIO2 (mean diurnal range), BIO1 (annual mean temperature), BIO9 (mean temperature of the driest quarter), BIO6 (min temperature of the coldest month), and BIO18 (precipitation of the warmest quarter). The average contribution rate of BIO2 in the eight models is 19.17% (Figure 3b), which is the highest of the five key bioclimatic variables, so it can be regarded as the most important bioclimatic variable. The total contribution rate of temperature-related key bioclimatic variables accounted for 65.87% ± 4.58% in each

model, and the total contribution rate of precipitation-related key bioclimatic variables was 14.74% ± 4.51%. Obviously, *M. glyptostroboides* is more sensitive to temperature than precipitation, which is conducive to further explore the response of *M. glyptostroboides* to the climate.

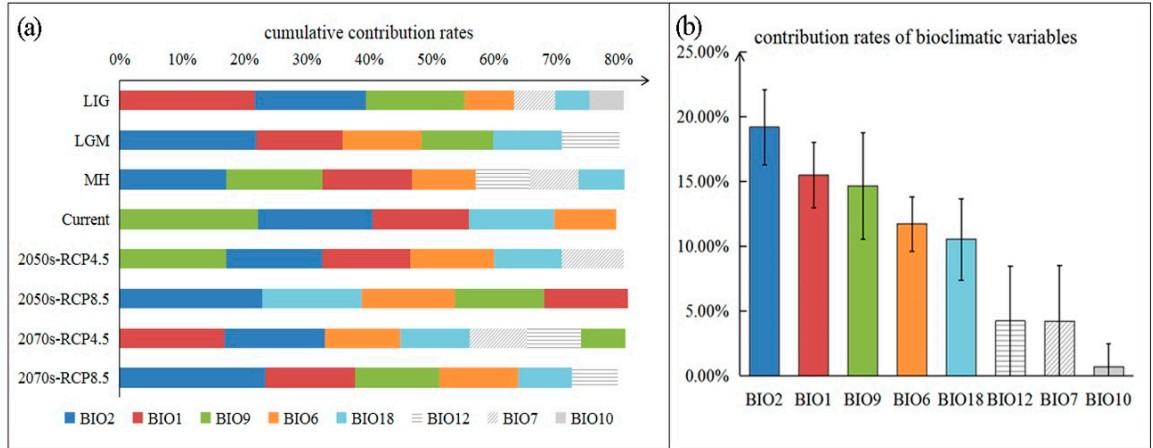

**Figure 3.** The key bioclimatic variables affecting the growth and distribution of *M. glyptostroboides*. (**a**) Statistics of the bioclimatic variables with cumulative contribution rates of more than 80% under each climate scenario. They are arranged by weight in descending order. Variables common to all scenarios are defined as key bioclimatic variables, which are shown in the colored (non-textured) portion. (**b**) The average contribution rate of each key bioclimatic variables in our models.

*3.2. Potential Distribution of M. glyptostroboides from the Past to the Future*

3.2.1. Potential Distribution of *M. glyptostroboides* in the Current Climate Scenario

Visually, relatively highly suitable habitats were concentrated in two areas (Figure 4d): (1) East Asia, mainly located in the North China Plain, the middle and lower reaches of the Yangtze River Plain, the Yunnan-Guizhou Plateaus in China, the Japanese archipelago, and the Korean Peninsula as well as (2) Western Europe, mainly located in the plains and a few mountainous areas. Moderately and marginally suitable habitats were scattered around the highly suitable habitats. Moreover, there are also some suitable habitats in the coastal areas of southeast Australia and some sporadic patches in America. Monsoons and circulations control and influence the climate in these areas. The current habitats with moderate and marginal suitability are approximately $4.698 \times 10^6$ km$^2$, and the highly suitable habitats cover approximately $2.159 \times 10^6$ km$^2$ (Figure 5).

3.2.2. Potential Distribution of *M. glyptostroboides* in the Past

In past climate scenarios, the potential distributions of *M. glyptostroboides* have undergone significant changes (Figure 4a–c). The total suitable area in the LIG was approximately $7.228 \times 10^6$ km$^2$, increased significantly to approximately $11.758 \times 10^6$ km$^2$ in the LGM, but decreased dramatically to approximately $7.807 \times 10^6$ km$^2$ in the MH (Figure 5). As we predicted, after a long historical vicissitude, the habitats of *M. glyptostroboides* seriously decreased in the MH. It is also a matter of concern that the range of the total suitable habitats of *M. glyptostroboides* has been decreasing since the MH period to the present, which means the species may be under environmental threat. Notably, the total number of suitable areas in the LGM showed a wider distribution compared with that of other scenarios (Figure 4b), which should be related to the relative maximum value of the total area of the continental shelf during this period [49].

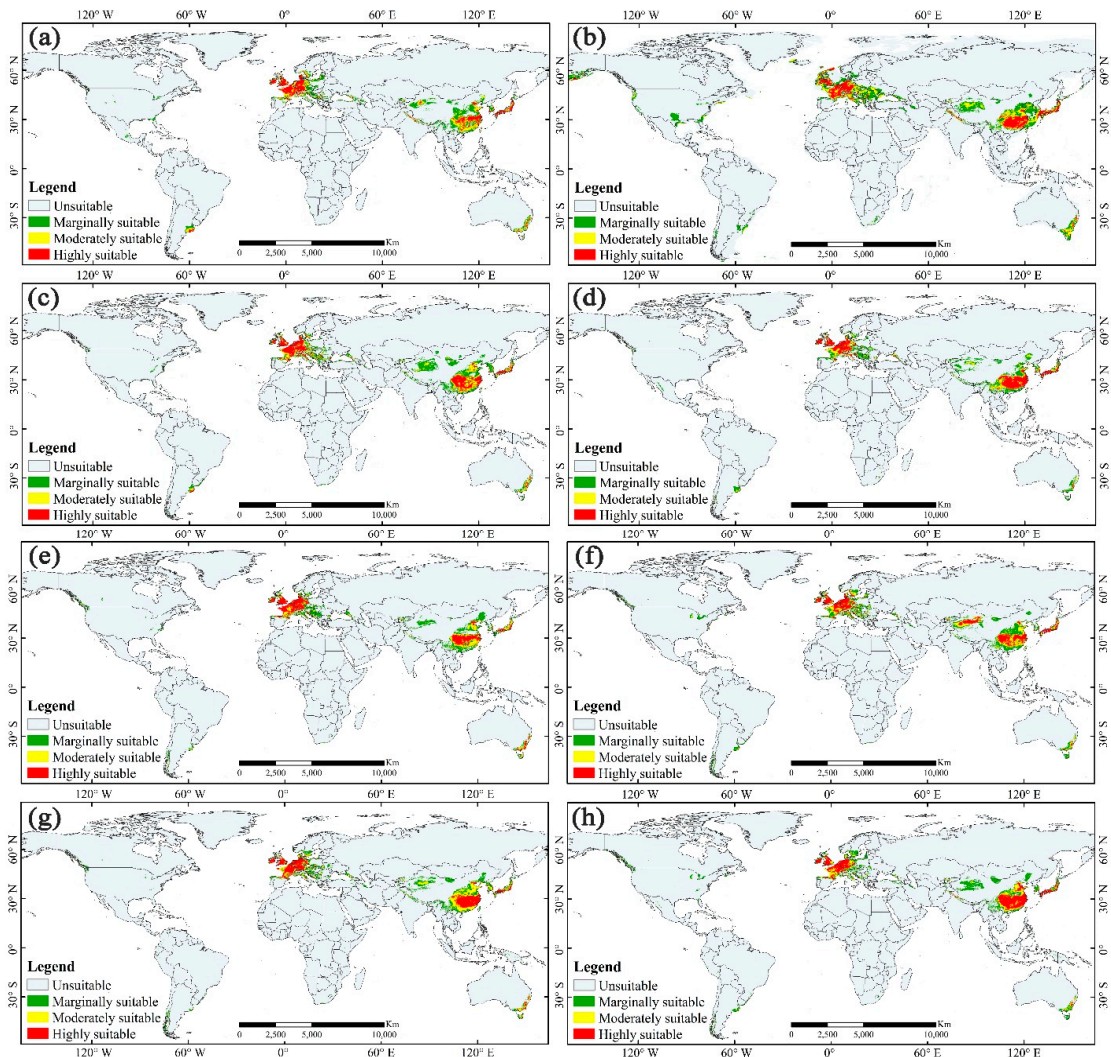

**Figure 4.** Potential distribution of *M. glyptostroboides* under each climate scenario. The green, yellow, and red represent the marginally suitable, moderately suitable, and highly suitable areas, respectively: (**a**) in the LIG, (**b**) in the LGM, (**c**) in the MH, (**d**) in the current period, (**e**) under RCP4.5 in the 2050s, (**f**) under RCP8.5 in the 2050s, (**g**) under RCP4.5 in the 2070s, and (**h**) under RCP8.5 in the 2070s.

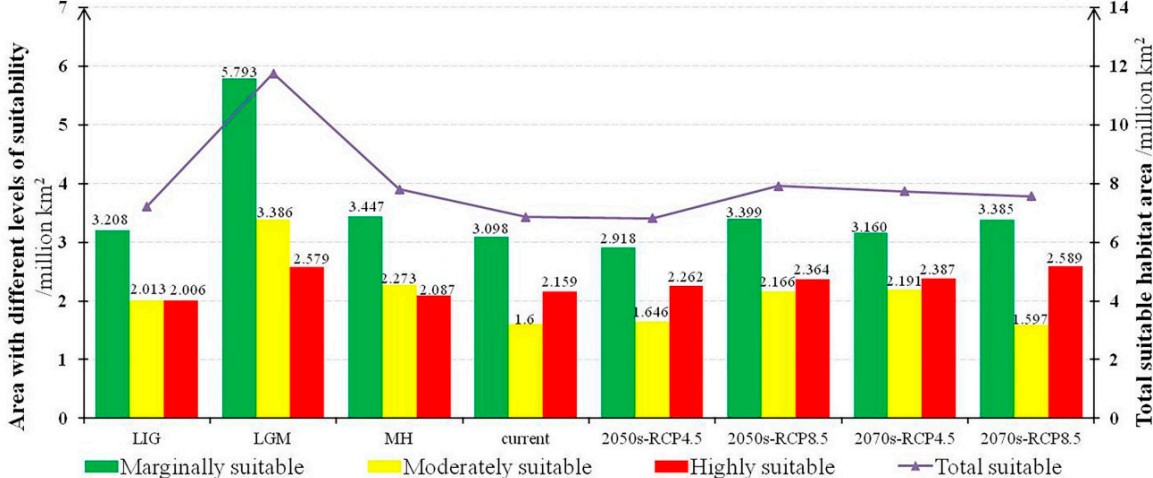

**Figure 5.** The total suitable habitat area and habitats with different levels of suitability under each climatic scenario. The details of the abbreviation in this figure are same as Figure 2.

In terms of the distribution details, we found that the potential distribution of *M. glyptostroboides* was more similar to the current distribution in Europe, while the significant change mainly occurred in Asia, especially in China. Taking the change of highly suitable habitat in China as an example, in the LIG, the highly suitable habitats of *M. glyptostroboides* were mainly distributed in the plain of the middle and lower reaches of the Yangtze River, and the habitat was narrow (Figure 4a). In the LGM, the highly suitable habitats of *M. glyptostroboides* expanded significantly, and moved southwest, even covering most of the Yunnan-Guizhou Plateau (Figure 4b). All habitats of *M. glyptostroboides* shrunk greatly in the MH (Figure 4c), resulting in the distribution area being highly reduced in China.

### 3.2.3. Potential Distribution of *M. glyptostroboides* in the Future

In future climate scenarios, the suitable habitats of *M. glyptostroboides* show an optimistic performance compared with the current scenario, which is characterized by a marked increase in suitable habitats as a whole (Figure 4e–h). The overall potential distribution will expand under climate change. Although the total suitable area may decrease slightly under RCP4.5 in the 2050s, it may increase significantly under other climate scenarios (Figure 5).

Under the scenario with a mild concentration of greenhouse gas emissions (RCP4.5), although the total suitable area would decrease by the 2050s, it would basically recover by the 2070s, and the area in the 2070s will still exceed the current area. The marginally suitable area would decrease by approximately $0.180 \times 10^6$ km$^2$ in the 2050s, but it would improve in the 2070s, and still increase by $0.062 \times 10^6$ km$^2$ compared with the current scenario. Moreover, with the increasing trend in suitable areas over time, by the 2070s, moderately and highly suitable areas will increase by $0.593 \times 10^6$ and $0.229 \times 10^6$ km$^2$, respectively. Thus, this slight reduction in the 2050s may even be considered unsurprising and not worth worrying about.

Under the higher emission scenario (RCP8.5), the potential distribution of *M. glyptostroboides* has a more optimistic trend with the increasing intensity of global warming. From now until the 2050s, the total suitable habitat area will increase conspicuously and will show a slight pullback by the 2070s, but will still be wider than the current area. In 2050, under RCP8.5, the overall suitable area of *M. glyptostroboides* will present the most optimistic state ($7.929 \times 10^6$ km$^2$), which is higher than under any other future scenarios (Figure 5). The discrepancy exists in the change trend in the suitable area from marginally to highly suitable levels, which would increase in the 2050s by $0.300 \times 10^6$ km$^2$, $0.568 \times 10^6$ km$^2$, and $0.206 \times 10^6$ km$^2$, respectively. From the 2050s to the 2070s, the marginally suitable area has a very small reduction, while the moderately suitable area decreases significantly by $0.569 \times 10^6$ km$^2$ in the 2070s. However, this small partial decline does not mean that the situation is dire. The highly suitable area would increase steadily, with an area of $0.225 \times 10^6$ km$^2$ more by the 2070s. In the 2070s, in addition to a negligible decrease in the moderately suitable area, the areas with marginal suitability and high suitability would be $0.286 \times 10^6$ km$^2$, and $0.431 \times 10^6$ km$^2$ more than the current area, respectively.

### 3.3. The Change in the Habitat Suitability of *M. glyptostroboides*

We analyzed the change in the habitat suitability of *M. glyptostroboides* in the near future climate scenarios by intersecting the level of HSI. Areas where the level of HSI will increase/decrease in the future climate scenario (compared to the current) are considered to be areas where the habitat suitability will definitely increase/decrease (Figure 6). Areas where habitat suitability will unchanged are also defined in the same way. In the context of global warming, the change in habitat suitability of *M. glyptostroboides* is not very pessimistic for the four future climate scenarios in our results. Overall, the suitability of about 40% of the habitats will be unchanged (Table 2). We expected that the suitability of *M. glyptostroboides* would be significantly decreased, especially in fragmented areas around the suitable area of the current scenario (Figure 6). However, the decrease is not as strong as the increase in all the scenarios. In terms of the whole habitat, the habitat suitability will increase to varying degrees in a wide range, especially in the core areas with high suitability. Under RCP4.5, the habitat suitability

will increase by 30.91% in the 2050s and by 37.49% in the 2070s. Under RCP8.5, the habitat suitability will increase by 38.68% in the 2050s and by 37.46% in the 2070s.

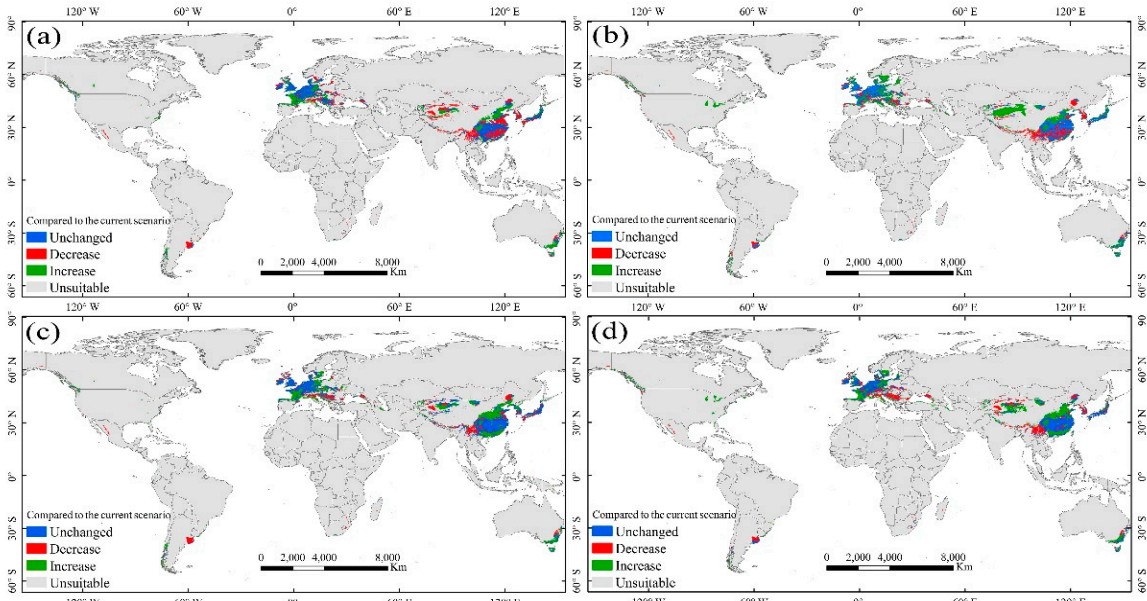

**Figure 6.** Trends in the habitat suitability of *M. glyptostroboides* under future climate scenarios (compared to the present day): (**a**) Under RCP4.5 in the 2050s, (**b**) under RCP8.5 in the 2050s, (**c**) under RCP4.5 in the 2070s, and (**d**) under RCP8.5 in the 2070s. The blue, red, and green represent the habitat suitability remaining unchanged, deceasing, and increasing, respectively.

**Table 2.** Statistics of the habitat suitability change in *M. glyptostroboides* in the future, which is expressed as the percentage of the habitat area with different trends in the total suitable area. The units are %.

| Climate Scenario | Decrease (%) | Increase (%) | Unchanged (%) |
|---|---|---|---|
| under RCP4.5 in the 2050s | 30.11 | 30.91 | 38.98 |
| under RCP4.5 in the 2070s | 18.98 | 37.49 | 43.53 |
| under RCP8.5 in the 2050s | 22.64 | 38.68 | 38.68 |
| under RCP8.5 in the 2070s | 22.68 | 37.46 | 39.86 |

### 3.4. Speculation on Glacial Refugia

We found proposed glacial refugia in East Asia and Western Europe on a large spatial scale. In addition, refugia may also exist in the coastal plains near the Gulf of Mexico and South-Eastern Australia (Figure 7). In East Asia, glacial refugia are most likely to exist in the following areas: the Tianshan Mountain, the Yunnan-Guizhou Plateau, the Sichuan Basin, the Qinling Mountains, the North China Plain, the plains in the middle and lower reaches of the Yangtze River, and the Nanling and Wuyi Mountains. In Western Europe, glacial refugia are mainly found in some plains and mountains (e.g., Western European plains, the Central Plateau, and the Alps). In order to explore the impact of future climate on these glacial refugia, we inferred the different situations of the refugia in the future by intersecting the future changes in habitat suitability of *M. glyptostroboides*. Regions of the refugia where the suitability will increase/decrease under future climate change are known as 'advantaged refugia'/'disadvantaged refugia', and the rest are called 'steady refugia'. The results showed that glacial refugia of *M. glyptostroboides* covers approximately $6.45 \times 10^6$ km$^2$, of which nearly 41.03% have a positive trend in habitat suitability, and 6.17% need to be treated with caution in the future.

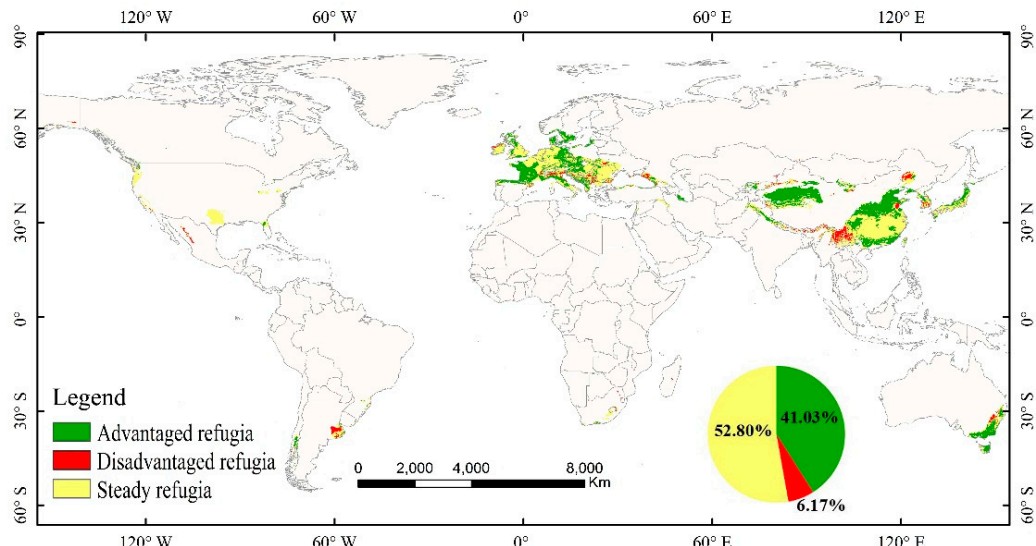

**Figure 7.** The location of glacial refugia of *M. glyptostroboides* and the changes in the refugia in future climate change.

### 3.5. Identification of Priority Protected Areas and Key Protected Areas

Based on the identifications of this study, the priority protected areas of *M. glyptostroboides* are mainly located in the piedmont of the Alps, Apennines, Dinara Mountains, and the plain of the middle reaches of the Danube River of Western Europe (Figure 8a), the Nanling Mountains, the Yunnan-Guizhou Plateau, the North China Plain, the Northeast China Plain of China, the piedmont of Taihang Mountains, and the Himalayas (Figure 8b). There are also a small number of other notable areas, such as the Japanese archipelago and the southern part of the Korean Peninsula (Figure 8b). The key protected areas are mainly located on the fragmented patches at the edges of potential distribution in the current scenario (Figure 8), such as the Taihang Mountains, Qinling Mountains, Sichuan Basin in Asia, and the northern shores of the Mediterranean and the southern shores of the Baltic Sea in Western Europe.

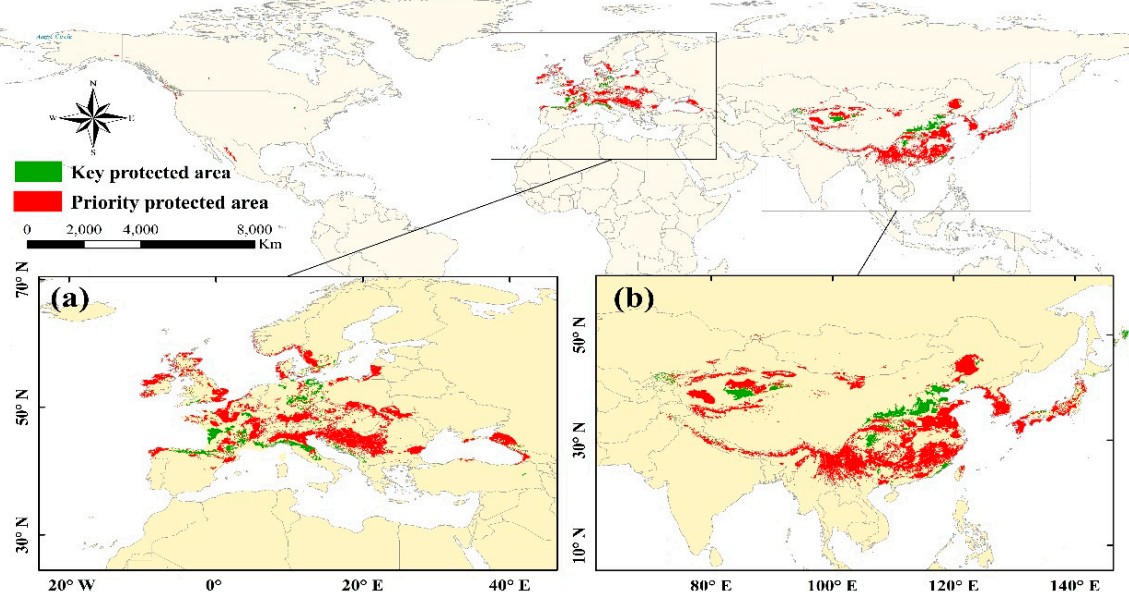

**Figure 8.** Identification of the priority protected areas and the key protected areas of *M. glyptostroboides*, which are mainly in Europe (**a**) and in East Asia (**b**).

## 4. Discussion

*4.1. The Rationality of the Model and the Limitations of the Prediction*

Mapping the potential distribution of *M. glyptostroboides* at a large spatial and temporal scale poses a unique challenge. SDMs provide a possible solution to this problem and already underpinned many biological conservation studies by providing comprehensive potential distribution maps [50–52]. Compared with traditional frequentist data models, machine learning models can provide obvious advantages to account for non-intuitive relationships and the anisotropy of ecosystems [53]. RF models have also shown to play an important role in exploring the effects of climate change [54]. In this study, the performance of the RF models is not only ensured by the powerful computational ability of the biomod2 package under the R environment, but the model accuracy is also improved through virtual distributions, cross-validation, and iterative algorithms. As expected, given the output results of the ROC, KAPPA, and TSS are all greater than 0.8, our model is considered to be trustworthy. Of course, the variance in a RF model should not be inflated, although it is not subject to over-fitting. It would be ideal to apply a combined model with less error to overcome many issues that need to be addressed in a single SDM framework [55,56].

The bioclimatic variables we used have substantial biological significance compared with single temperature or rainfall data and are collected from the most widely used dataset in the prediction of species potential distribution [57,58]. Although local factors (such as geographic location, soil, and topography) are sometimes also considered as important factors for maintaining botanic growth on a local scale, we assumed that bioclimate is the crucial driver of physiological processes related to species survival, because other factors are highly inconsistent and unavailable on large time scales [59,60]. Meanwhile, fully quantifying bioclimatic factors is difficult, and the obstacles of assessing anthropogenic influence would complicate the study. We could work in a more coordinated manner with experts in biogeography, ecology, and ecological physiology, since the dynamics and diversity of the species and biological interactions are not taken into account on the global scale [61]. Even so, the model framework and the bioclimatic variables used in this paper make the evaluation of the global potential distribution of *M. glyptostroboides* more accessible and acceptable.

In addition, consideration should be given to whether certain data from the botanical gardens can provide reliable indicators of species climate demand. As the species has been widely planted for ornamental purposes, it may not be practical to remove these suspect data. Thus, on the basis of obtaining a very limited natural distribution, the opportunities and limitations that some data from botanical gardens bring to climate-change research should also be discussed further. After the verification of the species records, the original species records from botanic gardens are within the expected suitable distributions. We considered that the distributions of plantation are also within the range of species suitable distribution, and the success of artificial cultivation of any tree species requires a long time to observe and verify whether the local climate is suitable for survival. Of course, the effects of microclimate and the use of irrigation on climate research that occurs in botanical gardens as well as ornamental conditions cannot be ignored. In further research, we can use the data from sufficient commercial forestry trials to evaluate broad scale suitability [62], or we can also play the role of botanical gardens by linking biodiversity conservation with the benefits derived from ecosystem services [63]. Organism–environment interactions at fine scales are important for assessing whether the microclimate is biologically relevant [64].

*4.2. The Significance of the Glacial Refugia Hypothesis*

Glacial refugia, where the climate is conducive to plant growth, are often considered as protectors of relict plants from extinction during long climatic cycles. The hypothesis of glacial refugia is important for further exploring the relationship between local species evolution and climate change. If we formulate strategies to explore the vitality of relict plants in the future, understanding which areas are conducive to the growth of *M. glyptostroboides* in the past is of utmost important. We assumed that the

climate of the glacial refugia can also provide suitable conditions for the growth of *M. glyptostroboides*, and it is also an essential reference for studying the survival of *M. glyptostroboides* after the Quaternary glaciar. A visual map was provided to help determine the location of refugia that may already exist but have not yet been discovered.

Our results showed that glaciation may play an important role in reducing genetic diversity, given that the distributions of *M. glyptostroboides* were significantly decreased during glaciation. We have determined that most of the glacial refugia are located in temperate regions with latitudes ranging from 20° N to 60° N in the Northern Hemisphere, where the climate is very suitable for *M. glyptostroboides* (the temperature and precipitation are almost the same as those in the LGM). Combined with the important research hotspot in temperate biodiversity [65], the results are deemed to be understandable. On the other hand, we tried to find real evidence that glacial refugia can exist in the mountains. During the last glacial period, temperate forests generated by sea level decline existed in the junction between eastern China, southern Japan, and the Korean Peninsula [66], which were conducive to the reproduction and survival of *M. glyptostroboides* at that time. After entering the interglacial period, these areas were gradually isolated and turned into mountains as temperatures rose, glaciers melted, and sea levels rose. Therefore, it is not surprising that the glacial refugia of *M. glyptostroboides* may exist in some high-altitude mountains.

In the past few decades, by using a species distribution model and systematic geographical surveys, some speculations about the location of glacial refugia have been confirmed, such as in Northern Europe [67], the mountains of Southern Europe (i.e., the Alps, Balkans, Pyrenees, and Apennines) [68], and Southeast Asia [69]. Phylogenetic analysis has also been used to support such hypotheses that some refugia were located in the Arctic or beyond during Pleistocene glaciation [70]. Previous studies have also shown in detail that the mountains of East Asia not only served as dispersal corridors, but also provided the refugia for some species during dramatic global climate change [71]. Three peninsulas (the Iberian, Italian, and the Balkan Peninsulas) of Europe are also traditionally considered to be glacial refugia. Although these places have been isolated for a long time, they can show the minimum systematic geographic structure to support plant growth [72]. Our results do not contradict previous studies, and can provide theoretical support for in-depth studies related to glacial refugia of relict plants.

Analyzing the applicability changes in future climate scenarios for glacial refugia will help us to protect the habitats of *M. glyptostroboides* in the next step. The predictions for the refugia will be better in the future, which gives us confidence in restoring *M. glyptostroboides*. In future research, paleoclimate data should be supplemented by the developing digital methods based on modern geography to overcome the error of inferring paleoclimate. Moreover, the delimitation of putative refugia requires the use of paleo-data and genetics in the further integration of climate models and biogeography, which encourages us to make progress in strengthening the collaboration between phytogeography and paleobiology.

### 4.3. Non-pessimistic Predictions of M. glyptostroboides under Climate Change Help Us Do More

*M. glyptostroboides* is not widely distributed at present and their suitable habitats will decrease partly under global warming. However, the good news is that the degradation of the habitats of *M. glyptostroboides* is not serious. We can see that most of the areas where the suitability would change are marginally suitable areas, while the moderately and highly suitable areas are basically unchanged and even increase (Figure 5). Moreover, the suitability of more habitats than expected will increase to varying degrees in future climate scenarios (Table 2). In the coming decades, large-scale global warming will dramatically influence the growth and the interaction of vegetation. Simultaneously, with the long-term and short-term changes of temperature, species will change in terms of population density, phenology, morphology, and genetic frequency, so the distribution of species will also change. Although rising temperatures have traditionally negative effects on ecosystems, such as increased population disease and the extinction of endangered species, climate change can also encourage

resilient species to adapt and thrive [73]. *M. glyptostroboides* has a strong ability to adapt to climate change, and will grow well in the climate conditions where temperature is relatively high but does not rise immoderately. This novel discovery at least suggests to us that *M. glyptostroboides* is almost no longer extinct or has been resurrected fully.

We reviewed the relevant bioclimatic variables and distributions of *M. glyptostroboides* under climate change, which will help to clarify the specific effects of climate change on the species, and to establish cost-effective strategies for conservation. As shown in our results, the temperature factors have a greater impact on *M. glyptostroboides* than the precipitations factors, which means that we have a direct motivation to explore the response mechanism between *M. glyptostroboides* and global temperature changes. By comparing the growth of *M. glyptostroboides* in different botanical gardens, some scholars also found that warm temperatures and temperature changes are necessary for the optimal growth of *M. glyptostroboides* [74]. By studying the general temperature-related physiological traits of species, species range was detected to be strongly dependent on temperature [75]. Therefore, it is acceptable to attribute predicted changes in the future distribution of plants to climate change, especially temperature changes. Prospective work could assess the trade-offs shown here between the growth of *M. glyptostroboides* and climate change to obtain an understanding of the ecosystem services of *M. glyptostroboides*, and even the contribution of nature to humanity.

Although these predictions indicated that the suitable habitats of *M. glyptostroboides* will expand and the suitability will increase, the stability and sustainability of its habitat cannot be maintained without prescient conservation. Currently, *M. glyptostroboides* has been widely grown in many countries, but the actual benefits of these suitable habitats are not ideal due to ecological destruction and urban development [76]. The native habitat of *M. glyptostroboides* is mostly disturbed by climate change and human factors and the frequency of *M. glyptostroboides* is decreasing in all directions from the center to the outside. This species is very sensitive to climate, meaning it is an indicator of climate change. Based on this, management actions for species conservation can be integrated, such as expanding the protected areas, maintaining or improving landscape connectivity, protecting the primary mother trees and genetic genes, and improving the microclimate by planting trees.

Conservation of species is not carried out evenly around the globe, but protected areas need to be identified. Therefore, we strongly recommend international research cooperation to protect *M. glyptostroboides* on a global scale. The establishment of the Dinghu Mountain Nature Reserve and the Xingdou Mountain Nature Reserve in China, for example, is making great contributions to the protection of *M. glyptostroboides*. However, existing nature reserves are in urgent need of improvement due to a lack of area and connectivity. It may be also feasible to combine the protected areas of *M. glyptostroboides* with nature reserves, which can protect the original parent tree and its habitats. We hope that identifying the protected areas in this study can also inspire the protection of *M. glyptostroboides*. The 'priority protected areas' that we have identified require further research in particular because of their vulnerability to the threat of climate change. Focusing on 'key protected areas' will facilitate the growth of species in future climatic conditions, which is considered to be the most cost-effective conservation strategy. Moreover, we can also consider other important impact metrics (e.g., abundance, range size, and species richness) to support conservation actions. In fact, the prediction of protected areas for relict plants is still in its infancy, and we hope that our studies will help researchers better assess these impacts, thereby making decisions to protect species and provide ecological services.

## 5. Conclusions

Based on the RF model, the potential distributions of *M. glyptostroboides* under multiple (the current one, the past three, and the next four) climate scenarios were predicted in this study. The key bioclimatic factors affecting the distribution of *M. glyptostroboides* are BIO1, BIO2, BIO9, BIO6, and BIO18, among which the importance of temperature factors is greater than precipitation factors. The relatively suitable habitats of *M. glyptostroboides* currently mainly distributed in East Asia and Western Europe, and its total area is approximately $6.857 \times 10^6$ km$^2$. Under past scenarios, the potential distributions of *M.*

*glyptostroboides* were not similar against time. We not only speculated the detailed locations of the refugia of *M. glyptostroboides*, but also predicted that glacial refugia are not pessimistic under climate change. Under future climate scenarios, the potential distributions of *M. glyptostroboides* have an optimistic trend in the context of global warming. Under the mild and higher emission scenarios, the total suitable habitat area in the 2070s would be more widespread than at present, and the level of the suitability of *M. glyptostroboides* will also show an obvious upward trend. For further long-term management and protected planning, the location of its priority protected areas and key protected areas were identified. These results can support the ongoing research on the interaction between climate change and *M. glyptostroboides* on a large space-time scale. We hope that our research will stimulate greater interest among policymakers towards protecting *M. glyptostroboides* and similar relict species.

**Author Contributions:** X.Z. (Xiaoyan Zhang) and W.G. designed and coordinated the research project, analyzed the results, and wrote the paper. W.G. and H.W. conducted formatting, and was the advisor and text editor. H.W. and X.Z. (Xuhui Zhang) participated in designing the models, analyzed the data. J.L. and Q.Z. gave comments and improved the final manuscript. All authors have read and agreed to the published version of the manuscript.

**Funding:** This research is supported by the National Natural Science Foundation of China (No. 31070293), and the Research and Development Program of Science and Technology of Shaanxi Province (No. 2014K14-01-02).

**Acknowledgments:** We are grateful to each of the resource sharing platforms, such as GBIF, ALA, CVH, WorldClim, and so on, which provide us with sampling data and climate data to be used in species distribution model. Also, we appreciate Yunfei Gu, a doctoral student at the University of Southampton for assisting us in revising the paper.

**Conflicts of Interest:** The authors declare no conflict of interest.

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
