# Peer review of "Non-Pessimistic Predictions of the Distributions and Suitability of Metasequoia glyptostroboides under Climate Change Using a Random Forest Model"

_forests, doi:10.3390/f11010062_

Round 1

Reviewer 1 Report

I am pleased to say, I have not revewed for a long time so well projected and conducted research. I have no serious remarks. I would only suggest to change the colour scale on result figs, puting the colours from green (low) through yelow (moderate) to red (high) suitability. This way the figs would more self explanatory. Congratulations!

Author Response

Thank you for your letter and comments concerning our manuscript entitled "Non-pessimistic predictions of Metasequoia glyptostroboides under climate change using random forest model " (ID: Forests-660669). We have studied the comments carefully and have changed the color scale on result figs according your suggestion. It can be seen in Figure 4 and Figure 5 in the revised manuscript. The green, yellow, and red represent the marginally suitable, moderately suitable, and highly suitable areas, respectively. We also hope that the figs would be more self explanatory in this way.

We greatly appreciate the efforts of the editor and reviewer, and we hope that the revised manuscript will meet with approval.

Once again, thank you very much for your comments and suggestions.

Looking forward to hearing from you soon.

With kindest regards!

Yours Sincerely

Xiaoyan Zhang, Haiyan Wei, and Wei Gu

Reviewer 2 Report

This is an intriguing paper which examines the distribution of Metasequoia glyptostroboides under past, present and future climatic conditions. It is a very unusual species having been rediscovered relatively recently (1941) in a very limited area and then very widely planted as an introduced species especially in Asia and Europe. This is in stark contrast to the data used in most previous SDM analyses of climate change impacts on forest species (see, for example, the 145 analyses listed in the supplement of Dyderski et al. 2018 , ref 58 here). Typically there are many observations from natural distributions and few (or none) from plantings outside the natural distribution. Unusually the analysis here predicts likely increases in habitat suitability under both mild (RCP4.5) and higher (RCP8.5) future climate change scenarios.

My main concern with the paper is that the data analysed here includes occurrences from botanic gardens and ornamental conditions where microclimate and the use of irrigation may mean that the interpolated climatic estimates are not reliable estimates of actual site conditions. Use of data from commercial forestry trials outside natural distributions tend to provide more reliable indicators of species climatic requirements as foresters are assessing broadscale suitability almost always without irrigation. For example, Pinus radiata is an endangered species in its very limited natural distribution of about 5000 ha in California, but the more than 4M ha of plantations particularly in the southern hemisphere provide useful information on its climatic adaptability. In contrast, much of the data used here comes from ornamental and/or botanic garden conditions both in Asia and Europe as well as outliers elsewhere. Analysis of these data may suggest climatic adaptability that M. glyptostroboides doesn’t have. This very important limitation should be considered in the discussion. As the species has been widely planted for ornamental purposes it may not be practical to remove these suspect data.

If the Editor desires a shorter paper consideration could be given to deleting sections 3.4 and 3.5 on glacial refugia and protected areas and related discussion. With regard to refugia is there any evidence (for instance from fossil records) that M. glyptostroboides ever existed in Europe? As M. glyptostroboides has been so widely and successfully grown ex-situ, conservation would seem a very low priority in comparison to many other species.

Aside from the concern about much of the distributional data coming from botanic garden and/or ornamental plantings the analysis methods are appropriate and the results are suitably presented. The text is generally good, but the English needs a little work e.g. delete 'sincerely' at line 451. 

In summary, it’s an interesting paper because of the unusual nature of the species (i.e. very limited natural distribution but very extensive plantings), but much of the data presumably coming from ornamental plantings is a potential problem.

Title – ‘Non-pessimistic’ is used in the title – why not use ‘optimistic’ as in the abstract? The word ‘distributions’ should appear in the title.

Abstract

Can we be sure the reduction has been due to climate change and not other factors such as competition with other species? Don’t the extensively climatically suitable areas shown in Fig. 4 suggest that factors other than climate were responsible for the drastic reduction in natural distribution? Line 21 Delete ‘glaciar’ and insert ‘glacial’ Line 30 Delete ‘total’ and insert ‘total area of’ Line 32 ‘within a wide range’ of what? Do you mean spatial range?

Introduction

L.42 You would expect a general text to be cited to support the idea that ‘climate is a crucial driver of physiological processes related to the survival of species’ not a very specific recent study. L.52 I suggest deleting the first sentence and writing ‘The first species distribution model (SDM) package called BIOCLIM was developed in the mid-1980s to study the effects of climate on species distributions (Booth et al. 2014).’ This reference is open access at https://onlinelibrary.wiley.com/doi/full/10.1111/ddi.12144 . It is relevant here as the authors are assessing data developed using thin plate spline climatic interpolation methods prepared initially for BIOCLIM in 1984 and a set of 10 variables selected from 19 developed for BIOCLIM in 1996. In addition, the effects of climate change on forest species distributions were first explored using SDMs in papers published in 1988 (but missed by Dyderski et al). I can’t blame the authors for missing these important facts as they are not described in the three SDM/ENM/HSM books or in the WORLDCLIM publications or website. L.71 Delete ‘a kind of’ and insert ‘an’ L.75 You need to add a sentence here to make it clear that the natural occurrence was extremely limited and almost all of the very extensive occurrences shown in Figure 1 are introductions. See https://www.conifers.org/cu/Metasequoia.php for info on distribution e.g. primary occurrence near Sichuan-Hubei border, but some outliers. It’s interesting that just one-three trees are thought to have provided the entire seed source for trees grown outside China before 1991. L.81 ‘In the last few decades, the comprehensive distribution of M. glyptostroboides has not been investigated..’. The numerous occurrences in Figure 1 suggest that the distribution has been assessed. You mean climatic factors influencing the distribution have not been investigated. L.122 The occurrences used for analysis include major centres in Europe and Asia, but also outliers in North America, South Africa and Australia. Inspection of some of these locations indicate that they include some botanic gardens. As mentioned above a problem with analysing rare and/or ornamental plants is that they may be grown under microclimate and/or irrigated conditions that do not represent conditions estimated by climatic interpolation.  

Author Response

Thank you for your letter and comments concerning our manuscript entitled "Non-pessimistic predictions of Metasequoia glyptostroboides under climate change using random forest model " (ID: Forests-660669). The comments were valuable for revising and improving our manuscript and had important guiding significance for our research. We have studied the comments carefully and have made revisions accordingly. According to your suggestion, to make the conclusion more persuasive, we added more background knowledge in the introduction, examined and optimized the results, and supplemented the discussions. The grammatical expression and sentence structure of this paper have been revised by Yunfei Gu, a doctoral student at Southampton University. We hope the revisions meet with approval. The major revised portions and added references were marked with red in the revision. Detailed explanations of the revisions are listed below point by point in the word.

We greatly appreciate the efforts of the editor and reviewer, and we hope that the revised manuscript will meet with approval.

Once again, thank you very much for your comments and suggestions.

Looking forward to hearing from you soon.

With kindest regards!

Yours Sincerely

Xiaoyan Zhang, Haiyan Wei, and Wei Gu
